# Consumers’ Segmentation Influences Acceptance and Preferences of Cheeses with Sanitary Inspection and Artisanal Seals

**DOI:** 10.3390/foods12203805

**Published:** 2023-10-17

**Authors:** Larissa Santos Pereira, Bruna Klein Borges de Moraes, Elizandro Max Borba, Bruna Bresolin Roldan, Rosiele Lappe Padilha, Voltaire Sant’Anna

**Affiliations:** 1Life and Environmental Area, State University of Rio Grande do Sul, Encantado Campus, Encantado 95960-000, RS, Brazil; larissa-pereira@uergs.edu.br (L.S.P.); bruna-klein@uergs.edu.br (B.K.B.d.M.); elizandro-borba@uergs.edu.br (E.M.B.); rosiele-lappe@uergs.edu.br (R.L.P.); 2EMATER/RS-ASCAR, Porto Alegre 90000-000, RS, Brazil; brunabre@gmail.com

**Keywords:** conjoint analysis, cluster, front of package, acceptance, artisan cheese, label

## Abstract

Food labeling serves as a crucial medium for industries to communicate product qualities to consumers. Sanitary inspection and artisanal seals are significant markers for traditional cheeses, yet current information on this topic is limited. Therefore, this study aims to evaluate the impact of sanitary inspection and the ARTE seal on the acceptance of artisanal cheese. To achieve this objective, four hypothetical cheese labels featuring all combinations of sanitary inspection and ARTE seals were presented to 404 consumers. These consumers rated their acceptance of each label, a conjoint analysis was conducted, and the relative importance of each seal was calculated. Subsequently, consumers were segmented using hierarchical cluster analysis. Their socio-demographic profiles were statistically correlated to the clusters through a chi-squared method. The results revealed the existence of three distinct consumer groups: those who strongly prefer cheeses with a sanitary seal (assigning a relative importance of 80.2% to the seal), those who favor cheeses with an artisanal seal (assigning a relative importance of 52.5% to the seal), and those for whom the presence of either seal did not significantly affect acceptance. Consumers residing in metropolitan areas generally placed less value on both seals, whereas frequent purchasers of artisanal foods and residents of rural areas showed a preference for the artisanal seal. Other socio-demographic variables did not statistically correlate with cluster membership. In conclusion, consumer segmentation based on preferences for sanitary inspection and artisanal seals in food labeling is vital for developing effective marketing strategies and food safety education policies.

## 1. Introductions

Food quality is shaped by consumers’ experiences, which are influenced by taste and quality cues that guide individual attitudes and behaviors at the point of purchase. Packaging serves as more than merely a protective layer for food; front-of-package labeling (FOP) has increasingly become a critical medium for communicating desired product characteristics to consumers. Extensive research has been conducted on information and quality logo certifications, as they serve as avenues for informing the public about health, sensory, and quality aspects, thereby enhancing acceptance, willingness to pay, and sensory expectations [1,2,3,4,5].

The demand for artisanal, handcrafted, and homemade items has recently witnessed significant growth in traditional foods. The market for such foods is expected to reach nearly USD 1.2 billion by 2027 [6,7]. Rivaroli et al. [8] conducted a comprehensive review of the scientific literature on artisanal foods and identified an information gap, particularly in developing countries and those in the southern hemisphere. Cheeses are widely consumed dairy products around the world, and in Brazil, artisanal cheeses hold historical, socioeconomic, and even environmental significance [9,10,11].

In 2018, the Brazilian government introduced and regulated the use of the ARTE seal, depicted in Figure 1. The objective of this seal is to promote and valorize traditional animal-origin foods produced in an artisanal manner [12,13]. The seal also aims to encourage small-scale producers to comply with inspection processes, which are considered crucial for preserving traditional products [10].

Additionally, current Brazilian legislation mandates the inclusion of a Federal, State, or Municipal seal on cheese labels, which is issued following sanitary inspection [12,13]. If a cheese factory undergoes municipal inspection, the product receives the Municipal Inspection Seal (*Selo de Inspeção Municipal*—SIM in Portuguese) (Figure 2).

The tension between traditional foods and governmental sanitary regulations is a global concern [14,15]. Hermann [16] noted that European regulations have emerged as non-tariff trade barriers for heritage foods from developing countries joining the European bloc, negatively impacting income generation and efforts to alleviate rural poverty. Waldmann and Ferr [17] explored American attitudes toward craft and unpasteurized cheeses, while de Freitas and Stedefeldt [15] argued that humanizing values can motivate food handlers to adopt safety practices. Nonetheless, artisanal cheeses have been consistently associated with foodborne microorganisms, posing a significant public health concern [11]. In this context, it is essential to consider consumers’ risk–benefit assessments, either consciously or unconsciously. For many, the benefits of food consumption—such as availability, cost, personal preferences, food quality, and sustainability—outweigh the associated risks [18].

Consumer behavior toward food purchasing is complex, making segmentation increasingly critical for accurately evaluating reactions to food choices. The current literature offers limited studies that assess the impact of artisanal and sanitary seals on front-of-package labeling (FOP), a vital medium for informing consumers about quality cues. Recently, Pereira et al. [5] found that such seals on Colonial cheeses—a cultural dairy product in Southern Brazil—impact sensory acceptance and emotions in consumers. However, a gap may exist between behavior and attitude, leading people to exhibit one form of purchasing behavior while holding different attitudes toward quality cues [5]. Therefore, examining the influence of quality seals on consumer acceptance and assessing whether segmentation plays a role in this behavior is an important step for both the food industry and agribusinesses. This will better inform the development of marketing strategies and general business operations. Accordingly, this study aims to evaluate the impact of sanitary inspection and the ARTE seal on the acceptance of artisanal cheeses and to correlate these preferences with consumers’ socio-demographic profiles statistically.

## 2. Literature Review

Quality labels on food front-of-package (FOP) labeling, such as those indicating origin, production method, and healthfulness, guide consumers in forming expectations about a product’s quality. Non-sensory cues like brand, Protected Designation of Origin (PDO), and Protected Geographical Indication (PGI) seals are considered highly useful factors when consumers are choosing between competitive products [4,19,20]. During the decision-making process related to purchasing, consumers seek information from both their memory and external sources. They process this data and store the outcomes of their purchase decisions for future reference. Consumers aim to synthesize this stored information when making subsequent decisions, indicating a quest for efficiency based on pre-existing knowledge [21].

Two general mechanisms assist people in categorizing products during the moment of food selection, enabling them to focus more intently on certain options. Bottom-up mechanisms, or data-driven processing, involve real-time information processing that occurs as individuals’ receptors take in new data. This approach does not rely on prior knowledge or experience and is driven by the importance of the stimulus itself. However, such an approach can be cognitively demanding. Alternatively, consumers often adopt economically prudent cognitive strategies, acting as “cognitive misers” [22,23]. They engage in a top-down approach, also known as category-driven processing, which involves interpreting new information based on pre-existing knowledge [22,23]. Over their lifetimes, consumers construct knowledge structures based on experiences, prior knowledge, emotions, and expectations. These schemas are then used to form hypotheses when faced with new choices, guiding product evaluations and informing inferences that extend beyond the presented information [22,23]. Importantly, these decision-making models are not mutually exclusive; consumers may switch between them depending on the situation and the product being considered.

Numerous studies have examined the influence of quality logos and other important cues on consumer acceptance and choice. Skubic et al. [20] evaluated the effect of origin (Slovenia, EU, and non-EU), label (national PDO and EU PDO), and price (EUR 12.50–20.00 EUR/kg). They observed that origin was the most critical factor for ham, particularly those from Slovenia. National PDO and PGI labels were more desired than products carrying the EU PDO and PGI labels. Segmentation showed that men more strongly preferred cheese without a label than women did. Women preferred the EU PDO-labelled cheese more strongly than men, and no significant differences were found in preferences among respondents based on education, employment, region, type of settlement, or income. In Germany, cheeses with geographical indication and organic labels impacted product prices more than the country of origin [24], although the country of origin was also an important factor. Schouteten et al. [25] evaluated the effects of labels like “with reduced salt,” “light cheese,” and “light and reduced salt” on cheese preferences. They observed that the “light” label led to lower overall expected and perceived liking for the same sample, the reduced salt label led to lower saltiness intensity, and the light label led to lower fat flavor intensity.

Dominick et al. [26] evaluated the “all natural” label across various products and observed that it positively impacted consumer behavior. Women were generally more receptive to the “all natural” label than men across all food categories studied. Ares et al. [1] observed in Uruguay that brand, price, and health claims significantly affected consumer choice for functional yogurts, with the relative importance of the brand being similar to that of the type of yogurt. Price was the most predominant factor. The authors also noted that consumer profiles could alter this effect, particularly their attitudes towards health, indicating numerous gaps yet to be explored, such as the influence of brand and packaging considering the emotions evoked and hedonic acceptance.

Emotions also play an important role in consumer food acceptance and are key tools for discriminating between foods. Schouteten et al. [25] observed that the emotional profiles of labeled cheeses (reduced salt and light) differed before tasting; however, few differences were found when actually tasting these cheeses. Regarding sanitary and ARTE seals, Pereira et al. [5] performed sensory acceptance tests in Brazil and found that cheeses without any seal on the front label had significantly higher acceptance than those with sanitary inspection or the ARTE seal. A sanitary inspection seal on cheese labels also evoked some negative emotions among Brazilian consumers, corroborating Miloradovic et al. [27], who observed that Serbian, Croatian, and Spanish consumers valued artisan cheeses more than industrial cheeses and had no clear opinion about the safety of artisan cheese. Most Brazilians claim to purchase dairy products without any sanitary inspection, mainly from short food supply chains such as street markets, cheese factories, or directly from producers [28,29]. Consumers of Canastra cheeses reported feelings of “happiness” and “pleasure” during home-use tests [30], and Colonial cheese has a positive reputation, offering a high cost–benefit ratio and meeting hygiene standards, thus satisfying both emotional and functional needs [31].

## 3. Materials and Methods

### 3.1. Sampling

Using snowball sampling, volunteers were recruited through social media advertisements, allowing participants to share the study link with others. This approach characterizes the study sampling as non-intentional and non-probabilistic. The sampling criteria included: (i) participants must be older than 18 years; (ii) reside in Brazil; and (iii) express an interest in participating. The advertisement outlined the study’s objectives, risks, and benefits. A minimum of 385 respondents was required, based on a 95% confidence level and a sample error margin *E* = 0.05. The proportion of favorable and unfavorable responses from an infinite population was considered to be 50% [32].

### 3.2. Survey and Product Design

The survey was administered through the Google Forms platform and received prior approval from the State University of Rio Grande do Sul Ethics Committee (Certificate of Presentation of Ethical Appreciation number 58043722.2.0000.8091). Data collection occurred between September and November 2022, beginning with a consent form for volunteer participation. After clicking on “I agree to participate”, respondents were able to proceed with the survey.

The independent variables, SIM and ARTE seals, each had two levels (with and without the seal) and were organized into a full factorial plan, resulting in four fictitious cheeses. This design was consistent with a prior study by Pereira et al. [5], which evaluated sensory acceptance of cheese samples and is now examining the impact of these seals on front-of-package labeling. Volunteers were randomly exposed to cheese labels (as determined by the software) and asked to rate their overall acceptance on a 7-point scale, ranging from “I disliked very much” to “I liked very much”. A pre-test involving five volunteers was conducted to ensure that the images were of good resolution, displayed correctly on both computers and mobile devices, and that the questions were clear.

In the second part of the survey, respondents answered socio-demographic questions covering gender, age, monthly income, education level, and city of residence. They were also asked about the frequency with which they purchase artisanal products. The regions were categorized according to the Brazilian Institute of Geography and Statistics, considering various cultural and historical influences [33].

### 3.3. Data Analysis

Data were organized using Microsoft Excel 2000 (MapInfo Corporation, Troy, NY, USA), and statistical analyses were conducted using XLSTAT software (Addinsoft, NY, USA, version 2022.3.1), Scilab (Rungis, France, version 6.1), or R software version 4.0.5. Conjoint analysis (CA) is a multivariate technique specifically designed to understand how consumers develop preferences for various types of objects. It allows for the evaluation of the influence of multiple attributes on consumer purchasing decisions [34]. Using Scilab software 2023.1.0, CA was performed on acceptance rates for each cheese label to calculate the coefficient of preference (CP) and importance (I) for each seal. The analysis followed multiple linear regression models as defined by the following equations:(1)$$Y_{jk}=\tau_{j}+\varepsilon_{jk}$$
where $\tau_{j}$ is defined as
(2)$$\tau_{j}=\beta_{0}+\sum_{i=1}^{m}\beta_{1i}X_{1i}^{j}+\sum_{i=1}^{m}\beta_{2i}X_{2i}^{j}+\cdots \sum_{i=1}^{m}\beta_{ni}X_{ni}^{j}$$

In these equations, *Y_jk_* is the acceptance rate to the *j*th treatment for the *k*th consumer; *β_si_* is the CP for the *i*th level of the *s*th factor; and *e_jk_* is the aleatory error associated with *Y_jk_*.

Consumer acceptance rates were centered around the average, and individual participant analysis was calculated as *Y_jk_* = *A_jk_* − *Ā._k_* and the data from all consumers (aggregated analysis), *Y_jk_* = *A_jk_* − *Ā*, where *A_jk_* is the consumer’s rating, *Ā._k_* is their average rating, and Ā is the general rating.

The model was then presented compactly in matrix notation as *Y* = *Xβ* + *ε*, where *Y* is the vector of observations of one or more consumers for the evaluated treatments, *X* is the matrix of indicator variables (indicating the presence—1—or absence—0—of factor levels) and *β* is the vector of CPs. The vector *β* was estimated by the ordinary least squares as follows:(3)$$\hat{\beta }={\left(X^{\prime }X\right)}^{-1}X^{\prime }Y$$
with the restriction of $\sum_{i=1}^{m}\beta_{si}=0$ for all factor *s*.

Then, the importance (I) was calculated as the CP amplitude:(4)$$I_{s}=Maximum({\hat{\beta }}_{si})-Minimum({\hat{\beta }}_{si})$$

The relative importance (RI) was calculated as
(5)$${RI}_{s}\left(\%\right)=\frac{I_{s}}{\sum_{s=1}^{m}I_{s}}$$

Using XLSTAT software, a hierarchical cluster analysis was carried out on the relative importance (RI) using Euclidean distances (for dissimilarity) and Ward’s aggregation criterion (as the agglomeration method). This approach helped identify clusters of consumers with different patterns. Within these clusters, the means of the CP and RI were compared using Analysis of Variance (ANOVA) followed by the Tukey test. Between-cluster comparisons were conducted using Student’s *t*-test in R software. Differences were considered statistically significant when *p* < 0.05.

To evaluate the statistical differences in volunteers’ acceptance rates for the cheeses (either for the entire sample or within clusters), tests for data normality and homogeneity of variances were performed using the Kolmogorov–Smirnov and Bartlett tests, respectively. The Box–Cox transformation was applied to ensure that the assumptions for ANOVA *p* > 0.05 for both the Kolmogorov–Smirnov and Bartlett tests were met. Acceptance means were then evaluated by two-way ANOVA, considering samples as fixed and consumers as random effects. This was followed by the Tukey test, with statistical differences deemed significant at *p* < 0.05. Principal Component Analysis (PCA) was performed on the acceptance data, using clusters as supplementary data in the XLSTAT software.

To examine the relationship between consumers’ socio-demographic profiles and the clusters to which they were assigned, the chi-square test was performed on a per-cell basis using the Fischer exact test. This aimed to identify any associations between socio-demographic profiles and the various categories. Statistical analyses were conducted using XLSTAT software, with a significance level set at 5%.

## 4. Results and Discussion

The socio-demographic profile of the 404 volunteers, which exceeded the minimum sampling number, is summarized in Table 1. The majority of participants were female (61.4%, *n* = 247), had completed high school (60.6%, *n* = 245), and earned three minimum wages or less per month (47.8%, *n* = 193). With regard to the frequency of purchasing artisanal products, 22.5% (*n* = 91) identified as frequent buyers, 38.6% (*n* = 156) as occasional buyers, and 38.9% (*n* = 157) as rare buyers.

Geographically, 51.2% (*n* = 207) of volunteers resided in metropolitan areas, while 48.8% (*n* = 197) lived in the countryside. The regional distribution showed that 61.3% (*n* = 247) were from southern Brazil, 14.1% (*n* = 57) from the southeast, 7.4% (*n* = 30) from the northern region, 8.4% (*n* = 34) from the northeast, and 8.7% (*n* = 35) from the midwestern region.

The results from the CA could not estimate the RI and CP for 107 volunteers, as they assigned the same score to all four labels to which they were exposed. Therefore, the presence of either the ARTE or SIM seals had no impact on the preferences of this group of consumers; this cluster was designated as G1. The means of acceptance rates, RI, and CP for each cluster are presented in Table 2.

For the remaining participants, hierarchical cluster analysis based on the calculated RI revealed two distinct consumer clusters. The first cluster, named G2, consisted of individuals with a strong preference for cheeses bearing the SIM seal (*n* = 121). For consumers in G2, the estimated RI for the SIM seal was 80.2 ± 14.2%, while for the ARTE seal it was 19.8 ± 14.2%. The second cluster, named G3, was composed of individuals who preferred cheeses with the ARTE seal (*n* = 177). For consumers in G3, the RI for the SIM seal was estimated at 45.2 ± 12.7%, and for the ARTE seal it was 52.5 ± 13.3%. The RI for the SIM seal was significantly higher in G2 compared to G3 (*p* < 0.05), while for the ARTE seal, the RI was significantly higher in G3 (*p* < 0.05).

Pereira et al. [5] observed that the presence of ARTE and/or SIM seals had a negative impact on the overall sensory acceptance of artisanal cheeses. Their findings align with those of the present study, which also identified groups that showed positive, negative, and neutral effects of front-of-package seal stimuli on the sensory acceptance of cheeses. Both G1 and G3 clusters were predominantly composed of women, while G2 was mainly men. Most participants in these clusters were in the age range of 31–40 years, earned between 1 and 3 minimum wages per month, and had completed college education. Further details on the profiles of these clusters are provided later in Section 4.

The valorization of artisanal foods is influenced by several factors [8]. Abouab and Gomez [35] reported that greater human involvement in food production leads to higher consumer confidence in the product’s craftsmanship and naturalness, as human processes are perceived to be more respectful of food integrity. Moreover, low technology utilization during production and high human involvement enhance consumer perception of the product’s craftsmanship [8]. Consumers also consider local foods to be more sustainable due to the shorter transportation chains, in addition to the importance of supporting local economies [36]. 

For CP, negative values signify that the level of the independent variable adversely affects acceptance, while positive values indicate a favorable impact. The absence of SIM and ARTE seals negatively influenced acceptance across both clusters. This finding is particularly striking considering that most Brazilian consumers typically purchase dairy products from open markets or directly from producers, often without encountering labels or official seals [28,29]. In contrast to this prevalent behavior, the results of the current study indicate that the lack of SIM and ARTE seals negatively affected respondents’ acceptance levels. Similar findings were reported by Sampalean et al. [19], who observed that for Italian consumers, the absence of a brand or PDO logo on Provolone cheeses led to negative values in conjoint analysis. These results suggest a disconnect between purchasing habits and attitudes, corroborating previous studies that identified a gap between consumers’ perception or knowledge and their attitudes toward food safety [18,37,38]. In alignment with the RI analyses, the absence of the SIM seal on the cheese label had a more detrimental impact on consumers in cluster G2 than those in G3 (*p* < 0.05). Conversely, the absence of the ARTE seal had the opposite effect. Likewise, a SIM seal on the dairy label had a more positive impact on consumers in G2 than in G3 (*p* < 0.05), whereas the opposite was true for the ARTE seal.

Consumers integrate various forms of information—about producer practices, social context, and the material qualities of a product—into an active, learned practice of sensory perception [39]. It is, therefore, critical to consider food consumption evaluations in their broader social context [40]. Schouteten et al. [25] found that Belgian consumers associated control-labeled cheeses with positive emotions such as gladness, happiness, and enthusiasm. Likewise, studies by Rodrigues et al. [30] and Steinbach et al. [29] found strong positive emotions connected to traditional cheeses produced in the southeast and southern regions of Brazil. However, segmentation remains a largely underexplored area, particularly concerning sanitary and artisanal aspects of food products. Adinolfi et al. [41] suggested that a designation of origin, while necessary, is insufficient for strong market performance. Cluster analysis appears to be an important tool for understanding consumer preferences. For instance, Mesías et al. [42] in Spain identified three groups of beef consumers, one of which placed significant importance on the product’s origin. Similarly, Sampalean et al. [19] segmented Italian volunteers into three clusters based on their preferences for Provolone cheese, finding that all preferred cheeses were produced by a national brand with an EU quality certification. Silva et al. [43] segmented Brazilian consumers based on their risk perceptions associated with cheese consumption. The authors observed that consumers with low perceived risk often cited sensory and quality aspects as the main factors influencing their purchasing decisions, whereas those with high perceived risk were more concerned with certification and labeling issues. Regarding acceptance, in cluster G1, the acceptance levels did not change across the various labels presented (*p* > 0.05). For clusters G2 and G3, labels without seals received the lowest acceptance scores (*p* < 0.05), indicating that the presence of these seals positively influences product acceptance. Specifically, consumers in G2 preferred cheeses with both the SIM and ARTE seals (*p* < 0.05). Conversely, consumers in G3 assigned the highest acceptance scores to cheeses bearing both seals (*p* < 0.05), with no significant difference between the ARTE and SIM seals (*p* > 0.05).

The analysis of acceptance rates across the clusters reveals distinct consumer preferences. Consumers in cluster G1 had the highest acceptance rates for cheeses without any seal, while those in G2 had the lowest (*p* < 0.05). In G1, the average acceptance score for this fictitious product was close to 6 on a 7-point scale, suggesting they “liked” the cheese label. In contrast, both G2 and G3 provided average scores lower than 4, indicating a general rejection of the cheese. For labels featuring the SIM seal, there was no significant difference in acceptance rates between G1 and G2 (*p* > 0.05), but G3 rated them the lowest (*p* < 0.05). The average score for products with the SIM seal was higher than 5, indicating a slight preference across all consumer groups. In the case of the ARTE seal, there was no significant difference in acceptance between G1 and G3 (*p* > 0.05), and both clusters seemed to like the product. However, G2 gave the lowest acceptance rates (*p* < 0.05), falling under a score of 4, which indicates that they neither liked nor disliked the product. When consumers were exposed to cheese labels featuring both the SIM and ARTE seals, no significant difference in acceptance was observed among the clusters (*p* > 0.05). The average scores were close to or higher than 6, suggesting a general liking for the product. Pereira et al. [5] previously noted various influences on sensory acceptance based on educational level and income, including more negative reactions to the ARTE seal among those with only a high school education. Figure 3 illustrates the results of the PCA, which explained 87.80% of the variance in the data. The first factor (F1) accounted for 66.57%, and the second factor (F2) explained 21.23%. As illustrated in Figure 3A, cheeses without any seal and those with the ARTE seal clustered closely together, as did those with the SIM seal and cheeses featuring both seals.

Analysis of Figure 3B shows that although there was a clear segmentation of volunteers based on their importance assigned to both seals evaluated, most consumers in G2 and G3 had similar profiles of acceptance rates (presented in the convergence of confidence ellipses of both groups), indicating that although they prefer the presence of a specific seal, they do not reject other. However, when samples and/or consumers were placed on opposite sides in PCA analysis, opposite behavior (for products) or rejection (for consumers) were indicated; thus, Figure 3B also shows the existence of niches of consumers that really prefer a seal and reject the other.

Consumers’ perception towards quality of foods has been changing and consumers’ behaviors and attitudes have been most influenced by cognitive processing of information and other quality aspects (such as price, affective issues, environmental aspects, or safety hazards), which have a large influence on the moment of purchasing [18,44]. Since the results indicated a clear segmentation of consumers, the independent chi-square test to evaluate consumers’ profiles was performed and the results are shown in Table 3.

Contrary to expectations, factors such as gender, age, monthly income, and educational level did not exhibit a significant relationship (*p* > 0.05) with consumer clusters. Similarly, the location where volunteers resided did not significantly influence their preferences for either the SIM or ARTE seals. This is noteworthy because previous research has shown that the region of residence can impact consumer perceptions, as evidenced in studies focusing on Brazilian consumers’ preferences for street market cheese (Silva et al., 2021) and Italian preferences for Provolone cheeses [19].

However, the frequency of purchasing artisanal foods did correlate with the segmentation (*p* < 0.05; degree of freedom [df] = 4; c^2^_observed_ = 9.610; c^2^_critical5%_ = 9.488). Frequent buyers of artisanal foods were predominantly associated with cluster G3, which showed a preference for the ARTE seal. Occasional and rare buyers did not strongly associate with any particular cluster. At a 10% confidence level, the place of residence also showed a statistical correlation with the clusters (*p*-value = 0.088; df = 2; c^2^_observed_ = 4.855; c^2^_critical10%_ = 4.605). Specifically, most consumers in cluster G3, who preferred the ARTE seal, resided in metropolitan regions, while those in cluster G1, who were indifferent to the presence of seals, primarily lived in rural areas. This is in line with research indicating that rural and urban residents have different consumer preferences regarding local foods [45,46,47].

Individuals residing in the countryside often have easier access to local food sources and tend to purchase dairy products directly from producers. Such products typically lack formal labeling, explaining the value that these consumers place on the presence of both the SIM and ARTE seals. Consequently, consumers in cluster G1 do not require FOP labels to verify a product’s artisanal status, as they usually source their products directly from the producer. In contrast, Borda et al. [48] found that approximately half of rural consumers, particularly younger individuals with lower- or middle-level education, lack awareness of biological hazards such as mycotoxins and pathogenic microorganisms.

Nonetheless, consumers in metropolitan areas often have fewer opportunities to buy directly from producers, accounting for their greater valuation of seals, indicating artisanal production. Additionally, with generally higher incomes, this demographic can afford the extra cost associated with products that carry the ARTE seal, which must be produced in industrial food processing plants instead of home kitchens where less investment in good manufacturing practices is required.

Multiple non-sensory factors influence consumer attitudes and behaviors and other dependent variables must also be considered. Attributes such as specific origin, type of cheese, production system, price, brand, and official PDO indications have been shown to impact consumer preferences and willingness to buy [19,49,50]. Furthermore, within the artisan food landscape, the type of product—cheese, ham, or honey—also plays a role [20,26]. Future research should extend these findings to a broader range of artisanal products to better understand these complex consumer behaviors.

## 5. Conclusions

In conclusion, this study underscores the importance of consumer segmentation in maximizing the positive effects of sanitary and artisanal seals on front-of-package labels for cheeses. The findings revealed three distinct consumer clusters: one that is not influenced by either seal and is primarily associated with rural living; a second that places high value on the presence of the sanitary inspection seal, irrespective of their socio-demographic profile; and a third that prefers the ARTE seal, mainly characterized by frequent purchases of artisan foods and residence in metropolitan areas. These insights contribute significantly to the existing literature and offer valuable guidance for professionals and regulatory bodies concerned with factors influencing consumer food acceptance, particularly when quality cues are added to FOP labels.

This study does have certain limitations. Firstly, it is a cross-sectional study, and attitudes may evolve, especially since the ARTE seal is relatively new in Brazil. Secondly, using a non-probabilistic sample could introduce bias; however, this was somewhat mitigated by recruiting respondents from various regions across Brazil, a country of continental dimensions with diverse cultures and profiles. Lastly, the study employed fictitious cheese labels, which could affect the overall perception of the cheeses in real-world scenarios. Future research should consider these limitations and explore other non-sensory attributes influencing consumer preferences, such as price, brand, and FOP color. Despite these limitations, the study’s intrinsic value and contributions to informing governmental policy on food safety and marketing in the cheese industry cannot be negated.

## Figures and Tables

**Figure 1 foods-12-03805-f001:**
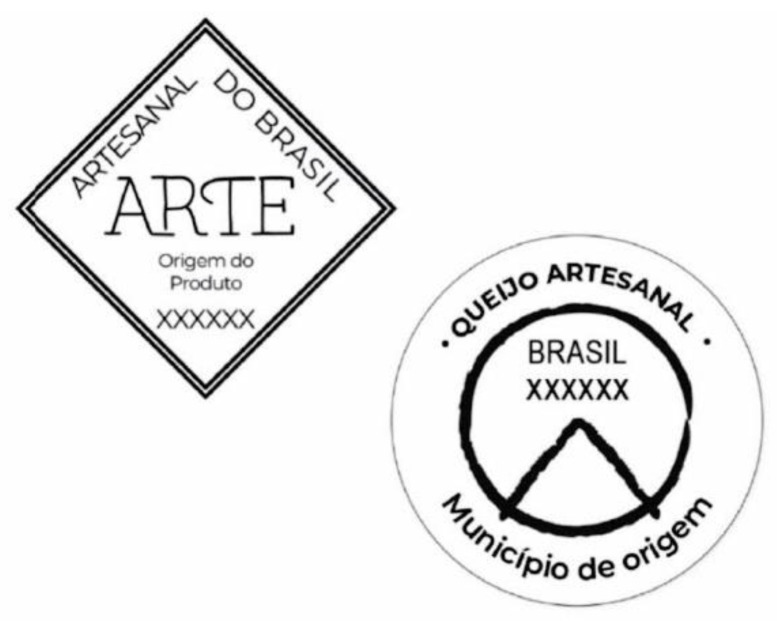
ARTE seal, according to Brazil [12,13].

**Figure 2 foods-12-03805-f002:**
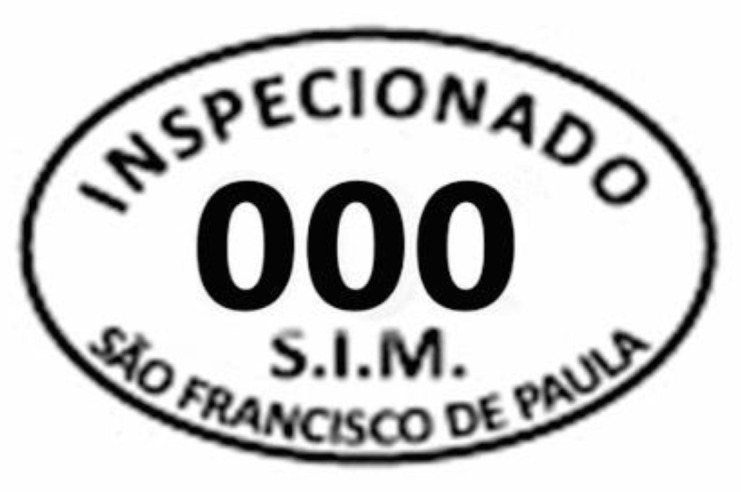
Municipal Sanitary Inspection Seal (SIM).

**Figure 3 foods-12-03805-f003:**
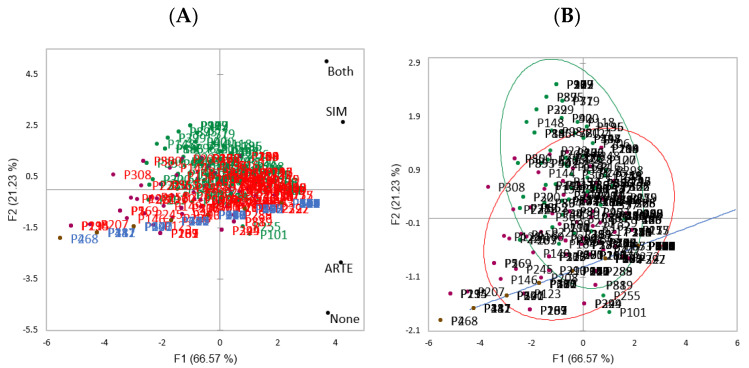
PCA analysis based on acceptance scores for the four fictitious cheese labels (**A**) and confidence ellipses (**B**) based on clusters. Consumers were colored by segmentation based on CA, and clusters were used as PCA supplementary material in the XLSTAT software to perform the confidence ellipses for cluster 1 (blue), cluster 2 (green), and cluster 3 (red).

**Table 1 foods-12-03805-t001:** Socio-demographic profile of the 404 volunteers.

Variables	*n*	f
Gender		
Male	155	38.6%
Female	247	61.4%
Age		
<20 years old	25	7.1%
21–30 years old	110	31.4%
31–40 years old	146	41.7%
41–50 years old	69	19.7%
>50 years old	54	15.4%
Level of education		
Incomplete fundamental school	6	1.5%
Complete fundamental school	22	5.4%
Complete high school	131	32.4%
Complete college	245	60.6%
Monthly income ^†^		
<1 minimum salary	56	13.9%
1–3 minimum salaries	137	33.9%
3–6 minimum salaries	76	18.8%
6–8 minimum salaries	48	11.9%
>8 minimum salaries	87	21.5%
Frequency of artisanal food purchases		
Frequent	91	22.5%
Occasional	156	38.6%
Rare	157	38.9%
Brazilian region		
Southern	247	61.3%
Southeastern	57	14.1%
Northern	30	7.4%
Northeastern	34	8.4%
Midwestern	35	8.7%
Area of residence		
Countryside	207	51.2%
Metropolitan	197	48.8%

^†^ Brazilian minimum salary in 2023 BRL 1380.00; USD 1.00 = BRL 5.15 on 8 October 2023.

**Table 2 foods-12-03805-t002:** Means and standard deviations of acceptance (based on a 7-point scale), relative importance, and preference coefficients of conjoint analysis for the different segmentation of SIM and ARTE seals.

	G1	G2	G3
	**Relative importance ^‡^**		
SIM	-	80.2 ± 14.2% ^Aa^	45.2 ± 12.7% ^Bb^
ARTE	-	19.8 ± 14.2% ^Bb^	52.5 ± 13.3% ^Aa^
	**Coefficient of preference ^†^**		
SIM_absence	-	−1.4 ± 0.76 ^Bd^	−0.55 ± 0.51 ^Ab^
SIM_presence	-	1.4 ± 0.76 ^Aa^	0.55 ± 0.51 ^Ba^
ARTE_absence	-	−0.3 ± 0.42 ^Ac^	−0.60 ± 0.47 ^Bb^
ARTE_presence	-	0.3 ± 0.42 ^Bb^	0.60 ± 0.47 ^Aa^
	**Acceptance ^†^**		
None	5.9 ± 1.6 ^Aa^	3.0 ± 1.6 ^Cc^	3.8 ± 1.7 ^Bc^
SIM	5.9 ± 1.6 ^Aa^	5.9 ± 1.1 ^Ab^	5.1 ± 1.4 ^bB^
ARTE	5.9 ± 1.6 ^Aa^	3.6 ± 1.7 ^Bc^	5.3 ± 1.3 ^Ab^
Both	5.9 ± 1.6 ^Aa^	6.4 ± 0.8 ^Aa^	6.1 ± 1.3 ^Aa^

Superscript capital letters indicate a significant difference between the columns (clusters) at 5% significance by the *t*-test. Superscript lowercase letters indicate a significant difference between lines at 5% significance by ANOVA followed by Tukey’s test (^†^) or the *t*-test (^‡^).

**Table 3 foods-12-03805-t003:** Contingency tables of clusters and consumers’ socio-demographic profile.

		G1	G2	G3	*p*-Value
Gender	Male	47	44	75	0.451
	Female	60	77	102	
Age	<20 years old	6	6	11	0.213
	21–30 years old	23	37	47	
	31–40 years old	37	40	67	
	41–50 years old	20	18	37	
	>50 years old	21	20	15	
Monthly income ^†^	<1 minimum salary	9	12	25	0.663
	1–3 minimum salaries	44	46	56	
	3–6 minimum salaries	20	23	29	
	6–8 minimum salaries	14	13	18	
	<8 minimum salaries	20	27	46	
Education level	Incomplete basic education	7	1	3	0.895
	Complete basic education	2	5	8	
	Complete high school education	37	37	61	
	Complete higher education	61	78	105	
Purchasing frequency of artisan foods	Frequent	29	32	26 *	0.048 *
	Occasional	42	41	73	
	Rare	36	48	78	
Brazilian region	Southern	68	77	103	0.237
	Southeastern	10	15	32	
	Northern	5	7	15	
	Northeastern	15	9	15	
	Midwestern	9	13	12	
Living location	Countryside	64 ^a^	62	82 ^a^	0.088 ^a^
	Metropolitan	43 ^a^	59	95 ^a^	

* *p* < 0.05; ^a^ *p* < 0.10; by chi-squared test per cell. ^†^ Minimum salary in Brazil in 2022: BRL = 1302.00. USD 1.00 = BRL 5.31 (18 December 2022).

## Data Availability

The experimental data obtained in the current study are available from the corresponding author on reasonable request.

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
