# Peer review of "Consumers’ Segmentation Influences Acceptance and Preferences of Cheeses with Sanitary Inspection and Artisanal Seals"

_foods, 2023, doi:10.3390/foods12203805_

Round 1

Reviewer 1 Report

The manuscript describes research conducted to explore consumer acceptance of cheese with sanitary inspection seal and artisanal seal. The objective of this study could be much clear and therefore will need some review.  The second part of the objective indicates analysis that was performed on the data obtained and will need rephrasing to reflect an appropriate objective. The title of the manuscript will also require reviewing to reflect the content of the manuscript. In my view, the segregation what should be taken out of the title to read ‘Consumer acceptance and preference of cheese with sanitary inspection and artisanal seals.

Abstract: will need reviewing, to include summary of data of the main findings of the study. It will be useful to present some summary of the data from the results. There is a grammatical error in line 12.

Similarly, there are some few grammatical errors throughout the manuscript that must be addressed by the authors.

Introduction: Introduction should be restructured for a better perception of the information by readers. There are some grammatical errors with some sections, with long sentences making them difficult to read and understand what authors wish to imply. E.g., Lines 94-95, 111-123, etc.

Something seem to be missing in line 76. It is also not clear if the reference cited in line 75 (Brazil, 2018, 2022) is appropriate. Is the author, Brazil, right? This reference citation could also be found in other parts of the manuscript.

Methodology:

Line 150: replace ‘advertising’ with advertisement’ and ‘by’ with ‘with’.

Line 157: replace ‘at’ with ‘on’.

Lines 161-163 will need to be rephrase for clarity.

Line 171: delete ‘structure’.

Line 174: replace ‘in’ with ‘on’.

Line 190: replace ‘at’ with ‘using’.

Line 193: delete ‘at’.

Line 194: delete ‘like indicated further’.

Line 198: replace ‘at’ with ‘using’.

Line 206: bsi is not presented in the equation indicated.

Line 224: replace ‘at’ with ‘using’.

Results and Discussion: There are few grammatical errors that will need addressing. Would it be useful to mention the 402 participants that were used (line 248) to correspond with the number provided in the methodology (line 152), instead of saying at least 385 respondents were needed. The question may also be asked; how many participants were recruited and how many responded. If this information is not available, I will suggest authors just mention the number of participants that were used in the study.

Lines 251-253: it is not clear what authors mean by ‘with complete high school’. Should this rather be ‘with high school qualification’ or participants who have completed high school? Similarly, what does ‘3 minimum salaries or less’ mean?

Line 257: replace ‘being’ with ‘with’.

Lines 258-260: There seem to be some confusion about the categorising of the regions: what is the difference between Southern Brazil and Southeast regions in Southern Brazil; in North region and Northeast? This can also be found in Table 1 and other part of the manuscripts. If there are differences, it would be beneficial for authors to provide this information in the methodology.

Line 275: IS ‘higher for G2 than G2’ right?

Line 276: Authors has cited Pereira et al (2023) as observing that globally the presence of ARTE and/or SIM seals impacted negatively on sensory acceptance of artisanal cheeses. The cited study was conducted in Southern Brazil; I am not sure what authors mean by ‘globally’. Authors must clarify this.

Lines 283-286: There seem to be a grammatical error, or the statement is incomplete. Please, rectify this for clarity.

Line 298: delete ‘no’.

Line 301-303: Rephrase for clarity.

Line 304, 305: replace ‘likewise’ with ‘similar to’; delete ‘to the’.

Lines 340-343: Statement seem incomplete.

Lines 345-351, 360-361, 370-373: confusing to read and understand. Will need rephrasing for clarity.

Line 413: replace ‘to’ with ‘with’. However, the enter paragraph (particularly Lines 413-419) is unclear and will need rephrasing for clarity.

Line 427: not clear what ‘to they being characterised’ mean. Check grammar.

Line 498: replace ‘adhere’ with ‘adhering’.

Lines 504-508: Check grammar.

Line 511: ‘for to’? check.

Authors to check and ensure all references are listed according to the journal’s requirements.

Lines 567-568: check reference: Authors, title (is it a book? Journal? Website information?)

References number 14 and 15 will require reviewing for appropriate listing of authors.

There are few grammatical errors to be addressed. Some sentences are unclear and would require rephrasing for clarity.

Author Response

Reviewer #1

The manuscript describes research conducted to explore consumer acceptance of cheese with sanitary inspection seal and artisanal seal. The objective of this study could be much clear and therefore will need some review. The second part of the objective indicates analysis that was performed on the data obtained and will need rephrasing to reflect an appropriate objective. The title of the manuscript will also require reviewing to reflect the content of the manuscript. In my view, the segregation what should be taken out of the title to read ‘Consumer acceptance and preference of cheese with sanitary inspection and artisanal seals.

Abstract: will need reviewing, to include summary of data of the main findings of the study. It will be useful to present some summary of the data from the results. There is a grammatical error in line 12.

Thank you for your observations and suggestions. The abstract was proofread and additional data were added.

Similarly, there are some few grammatical errors throughout the manuscript that must be addressed by the authors.

The entire manuscript was revamped and proofread by a professional language editing company.

Introduction: Introduction should be restructured for a better perception of the information by readers. There are some grammatical errors with some sections, with long sentences making them difficult to read and understand what authors wish to imply. E.g., Lines 94-95, 111-123, etc.

The introduction was shortened to contextualize the main problem in the manuscript, and a section of literature review was added with more information on the topic.

Something seem to be missing in line 76. It is also not clear if the reference cited in line 75 (Brazil, 2018, 2022) is appropriate. Is the author, Brazil, right? This reference citation could also be found in other parts of the manuscript.

The reference Brazil is the law where the seals are published. It is those evaluated in the present work and others from around the world to indicate artisanal production officially

Methodology:

Line 150: replace ‘advertising’ with advertisement’ and ‘by’ with ‘with’.

This correction was applied.

Line 157: replace ‘at’ with ‘on’.

This correction was applied.

Lines 161-163 will need to be rephrase for clarity.

The sentence was rephrased.

Line 171: delete ‘structure’.

The suggestion was applied.

Line 174: replace ‘in’ with ‘on’.

The preposition was fixed.

Line 190: replace ‘at’ with ‘using’.

The suggestion was applied.

Line 193: delete ‘at’.

 The suggestion was accepted.

Line 194: delete ‘like indicated further’.

Thank you for the suggestion. The sentence was deleted.

Line 198: replace ‘at’ with ‘using’.

 The suggestion was accepted.

Line 206: bsi is not presented in the equation indicated.

In Equation 2, s is the sth factor while i is the level. So, for example, β1i 1 is the first level and i is the factor. The equation was the same as presented in other works to evaluate Conjoint analysis data.

Line 224: replace ‘at’ with ‘using’.

 The word was changed.

Results and Discussion: There are few grammatical errors that will need addressing. Would it be useful to mention the 402 participants that were used (line 248) to correspond with the number provided in the methodology (line 152), instead of saying at least 385 respondents were needed. The question may also be asked; how many participants were recruited and how many responded. If this information is not available, I will suggest authors just mention the number of participants that were used in the study.

Thank you for the suggestion. It was added to the beginning of the section.

Lines 251-253: it is not clear what authors mean by ‘with complete high school’. Should this rather be ‘with high school qualification’ or participants who have completed high school? Similarly, what does ‘3 minimum salaries or less’ mean?

 This information was corrected.

Line 257: replace ‘being’ with ‘with’.

 The word was corrected.

Lines 258-260: There seem to be some confusion about the categorising of the regions: what is the difference between Southern Brazil and Southeast regions in Southern Brazil; in North region and Northeast? This can also be found in Table 1 and other part of the manuscripts. If there are differences, it would be beneficial for authors to provide this information in the methodology.

Thank you for the suggestion. The information was added.

Line 275: IS ‘higher for G2 than G2’ right?

 The information was corrected.

Line 276: Authors has cited Pereira et al (2023) as observing that globally the presence of ARTE and/or SIM seals impacted negatively on sensory acceptance of artisanal cheeses. The cited study was conducted in Southern Brazil; I am not sure what authors mean by ‘globally’. Authors must clarify this.

 The information was corrected. Actually, we meant the overall sensory acceptance.

Lines 283-286: There seem to be a grammatical error, or the statement is incomplete. Please, rectify this for clarity.

 The sentence was corrected.

Line 298: delete ‘no’.

 The sentence was corrected.

Line 301-303: Rephrase for clarity.

 The sentence was rephrased.

Line 304, 305: replace ‘likewise’ with ‘similar to’; delete ‘to the’.

 The sentence was changed.

Lines 340-343: Statement seem incomplete.

 The sentence was changed.

Lines 345-351, 360-361, 370-373: confusing to read and understand. Will need rephrasing for clarity.

 The sentences were changed.

Line 413: replace ‘to’ with ‘with’. However, the enter paragraph (particularly Lines 413-419) is unclear and will need rephrasing for clarity.

 The sentence was changed.

Line 427: not clear what ‘to they being characterised’ mean. Check grammar.

 The sentence was rephrased

Line 498: replace ‘adhere’ with ‘adhering’.

 The spelling was corrected.

Lines 504-508: Check grammar.

 The sentence was rephrased.

Line 511: ‘for to’? check.

 The sentence was corrected.

Authors to check and ensure all references are listed according to the journal’s requirements.

According to the Journal’s guidelines, the references can be formatted freely. Thus, I kept the reference style to make the whole process quick.

Lines 567-568: check reference: Authors, title (is it a book? Journal? Website information?)

According to the Journal’s guidelines, the references can be formatted freely. Thus, I kept the reference style to make the whole process quick.

References number 14 and 15 will require reviewing for appropriate listing of authors.

 The references were reviewed.

Reviewer 2 Report

Thank you for the opportunity to review your paper. I appreciate it. Please find below my comments who are hopefully helpful

1) Line 125-142: Please clearly emphasize what is the originality, novelty and merit of your paper. Certification and labeling is extensively studied.

2)Please also emphasize where within theory branches in extant literature your work is grounded.

3) Can the authors please justify why the authors conducted a power analysis for a convinience sample? Please explain the appropiateness given the sample is not representative of the population. Given you followed a snowball sampling, please explain how you save-guarded against bias.

4) Please elabrorate how many people were recruited and indicate a drop- out rate. 

5) Can the authors be more transparent on the methodological design of their work. Line 167 -169 reads like your experiential design is grounded in the literature. But that is your own work. That needs be made more transparent. 

6) Can the authors elaborate in their method section about choice experiments and conjoint analysis. 

7) In the conclusion please elaborate on the theoretical value of your work. Strenghen the recommandation for practitioners and add suggestions for future studies

Author Response

Thank you for the opportunity to review your paper. I appreciate it. Please find below my comments who are hopefully helpful

1) Line 125-142: Please clearly emphasize what is the originality, novelty and merit of your paper. Certification and labeling is extensively studied.

Although certification seals have been largely evaluated, sanitary inspection and artisanal seals have been little explored. Additionally, we bring information on the effect of consumers’ profiles on their acceptance, which is brand-new information to the current literature. Also, Rivaroli et al. (2020) conducted a wide range review of scientific articles about artisanal foods and noted a lack of information in developing countries, mainly in southern hemisphere countries. Looking for a holistic view of different cultures about this important issue, the present work brings important information to contribute to the scientific data about consumers behaviors that affect food acceptance. More detailed information was added in the text to reinforce this.

2)Please also emphasize where within theory branches in extant literature your work is grounded.

Thank you for the comment. A literature review was added to the manuscript.

3) Can the authors please justify why the authors conducted a power analysis for a convinience sample? Please explain the appropiateness given the sample is not representative of the population. Given you followed a snowball sampling, please explain how you save-guarded against bias.

We collected data from all over the country, and results showed that they present different behaviors toward acceptance of the variables studied. Although it does not represent the whole Brazilian population, the behaviors found in the results are in line with some findings in the literature.

4) Please elabrorate how many people were recruited and indicate a drop- out rate. 

Thank you for the comment. Unfortunately, we do not have this number (of people who received the recruitment advertisement or saw it on social media).

5) Can the authors be more transparent on the methodological design of their work. Line 167 -169 reads like your experiential design is grounded in the literature. But that is your own work. That needs be made more transparent.

Indeed, Pereira et al. (2023) is a work we performed before and published in the Journal of Sensory Studies. In this previous work, we got data about consumers’ acceptance of cheese after they eat it. In the present work, we are evaluating the acceptance of the seal on the front of the packaging. The results bring interesting information since consumers liked the seals on FOP, but our previous work showed that they reduce sensory acceptance. Data from both studies show a gap between food choice and overall food acceptance for both seals, which is critical information to understand factors that influence consumers’ behavior and attitudes towards foods. We clarified this in the text.

6) Can the authors elaborate in their method section about choice experiments and conjoint analysis. 

Although we are aware of the choice-based conjoint analysis, we chose the acceptance rate since we also sought to understand if consumers reject the seals. In the preference methodology, if a volunteer chooses a sample, it does not mean that they rejected the other. Thus, as a hypothesis worked before the beginning of the data collection, we decided to ask for acceptance and not choice. Unfortunately, the data we collect do not allow us to work now with choice-based analysis.

7) In the conclusion please elaborate on the theoretical value of your work. Strenghen the recommandation for practitioners and add suggestions for future studies

Thank you for the suggestions. The conclusion section was improved.

Reviewer 3 Report

General comments:

More prominent results should be included in the abstract. The abstract contains descriptive and to some extent minor results

The introduction should be modified to improve the readability. It contains many vague and long phrases

The conclusion should be improved. It contains many conclusions that are not related to the results of the study. Moreover, it is very long and narrative.

Line 34-37: The phrase is vague. It should be restructured

Line 45: “Quality labels on food FOP”. I think there is no need for FOP, it is a type of repetition

Line 72-78: it is very long phrase. It needs fragmentation to impart clear meaning

Line 94-95: The phrase is vague. It should be restructured

Line 115: poverty alleviation;” it should be ended by full stop.

Line 115-116: The phrase is vague. It should be restructured

Line 116: The phrase should be ended by full stop.

Line 117-118: The phrase is vague. It should be restructured

Line 135-138: The phrase is vague. It should be restructured

Line 249: Silva et al. (2021), this citation was not in the context of the results

Line 262-263: “The results from CA did not allow to estimate RI and CP for 107 volunteers”, Why?

Line 275: “RI for SIM seal was higher for G2 than G2 (p<0.05), meanwhile for ARTE seal RI was higher 276 for G3 (p><0.05)” How G2 is higher than G2.

278-280: authors indicate the conditions of groups that presented positive, negative and neutral effect of the front-of-package seal stimuli

Line 346: chesses, should be corrected

Authors should define figure 3 A and Figure 3 B

Line 413: in should be in normal font not in bold

Line 492-501: it is not conclusion related directly to the findings of the study

English language should be improved.

Author Response

More prominent results should be included in the abstract. The abstract contains descriptive and to some extent minor results

Thank you for the comment. The abstract was proofread and more details were added.

The introduction should be modified to improve the readability. It contains many vague and long phrases

The introduction section was shortened to contextualize the main problem in the results and a literature review was added to ground the issue.

The conclusion should be improved. It contains many conclusions that are not related to the results of the study. Moreover, it is very long and narrative.

 The conclusion section was reorganized.

 Line 34-37: The phrase is vague. It should be restructured

 The sentence was rephrased.

Line 45: “Quality labels on food FOP”. I think there is no need for FOP, it is a type of repetition

 The sentence was rephrased.

Line 72-78: it is very long phrase. It needs fragmentation to impart clear meaning

 The sentence was rephrased.

Line 94-95: The phrase is vague. It should be restructured

 The sentence was rephrased.

Line 115: poverty alleviation;” it should be ended by full stop.

 The sentence was rephrased.

Line 115-116: The phrase is vague. It should be restructured

 The sentence was rephrased.

Line 116: The phrase should be ended by full stop.

 The sentence was rephrased.

Line 117-118: The phrase is vague. It should be restructured

 The sentence was rephrased.

Line 135-138: The phrase is vague. It should be restructured

 The sentence was rephrased.

Line 249: Silva et al. (2021), this citation was not in the context of the results

 The sentence was deleted.

Line 262-263: “The results from CA did not allow to estimate RI and CP for 107 volunteers”, Why?

Thank you for the comment. When the volunteers give the same score for all the samples presented, the increasing rate is zero, and the relative importance is zero (since the presence of seals did not impact acceptance). Thus, it is not possible to calculate the parameters.

Line 275: “RI for SIM seal was higher for G2 than G2 (p<0.05), meanwhile for ARTE seal RI was higher 276 for G3 (p><0.05)” How G2 is higher than G2.

 The information was fixed.

278-280: authors indicate the conditions of groups that presented positive, negative and neutral effect of the front-of-package seal stimuli

A brief information was given, and a detailed discussion was presented in Table 3.

Line 346: chesses, should be corrected

 It was corrected.

Authors should define figure 3 A and Figure 3 B

 The figure caption was fixed.

Line 413: in should be in normal font not in bold

 The sentence was fixed.

Line 492-501: it is not conclusion related directly to the findings of the study

 The conclusions section was rewritten.